# Single-Shunt Measurement of Three-Phase Currents for a Three-Level Inverter under the Low Modulation Index Operation

**DOI:** 10.3390/s22062249

**Published:** 2022-03-14

**Authors:** Haris Kovačević, Dušan Drevenšek, Željko Ban, Rajko Svečko, Miro Milanovič

**Affiliations:** 1Emsiso d.o.o., 2211 Pesnica pri Mariboru, Slovenia; hariskovacevi@gmail.com; 2Mahle Electric Drives Slovenija d.o.o., 5290 Šempeter pri Gorici, Slovenia; dusan.drevensek@mahle.com; 3Faculty of Electrical Engineering and Computing, University of Zagreb, Unska 3, 10000 Zagreb, Croatia; zeljko.ban@fer.hr; 4Faculty of Electrical Engineering and Computer Science, University of Maribor, Koroška Cesta 46, 2000 Maribor, Slovenia; rajko.sveckoc@um.si

**Keywords:** power electronics, current reconstruction, single-shunt, three-level inverter, vector injection

## Abstract

This paper deals with the three-phase current reconstruction method under the low modulation index operation of three-phase three-level PWM inverters by using the single-shunt current signal measurement. The shunt is placed in a DC-link. The proposed reconstruction method is based on the combination of collinear vector injection and shifting of Space Vector Modulation (SVM) signals. The method offers a good solution for the area where the small modulation index appears. In this case, a lack of measurement time exists due to the narrow triggering pulses. This approach was studied theoretically and verified by laboratory experiments.

## 1. Introduction

This paper deals with the current reconstruction method for three-level inverters by using the single-shunt measurement placed in the inverter DC-bus. Such an approach is the subject of researchers’ interest because it enables reducing the number of sensors and can also be explored in fault-tolerant systems where safety is the main issue. Single-shunt current reconstruction under the low modulation index operating area is the subject of researchers’ interest for the applications where the permanent magnet synchronous motor is operating inside the low-speed region [1]. For such an operation, space vector modulation using a shift approach [2] cannot be applied as it causes the increase of the current ripple, which then can lead to the current reconstruction errors.

A different current reconstruction method was studied for two-level three-phase inverters in [3]. The authors suggested the current reconstruction strategy based on online offset compensation. This method is applied for two-level three-phase inverters, and it is also appropriate for operations with a low modulation index. In [4], the authors discussed the single-shunt measurement approach in order to reconstruct the three-phase inverter current for a hardware-in-the-loop system where the ultra-low latency of signals is required. The work performed by [5,6] dealt with exploiting the single-shunt measurement reconstruction and three-phase current measurement for safety issues. Different aspects of using the single-shunt reconstruction method for three-phase inverters were discussed in [7,8,9], where the authors studied the zero voltage sampling method and three-phase current reconstruction using three shunts placed in the collectors of bottom inverter transistors, respectively. The single-shunt measurement is applied in many works considering fault diagnoses and motor control, as in [10,11,12,13,14,15,16,17]. The use of a single Hall-principle-based sensor placed in a DC-link for three-phase current reconstruction and short-circuit identification was described in [18].

Some research has been performed on three-level and/or multi-level inverters. An interesting work was performed in [19], where the authors described an application where the three-level inverter was used for driving a five-phase electrical motor. However, the three-level (or multi-level) converters have recently been the subject of research work due to their good properties (voltage stress on the transistors), where the authors studied the control and design of the motor drives and PFC rectifiers [20,21,22,23]. In [24], a voltage injection method was proposed to solve problems with single-shunt current measurement applied to the neutral-point-clamped three-level inverter, which relocates the original voltage reference in the boundary area to the normal operating area in the voltage plane.

The SVM method for low modulation index values is studied and applied to the three-level inverter shown in Figure 1. The authors in [25] described the shifting of the SVM signals to ensure enough time for current measurement, which cannot be applied to the low modulation index operating area since the duty cycles for all three phases are similar and around 50%. The three-phase currents’ reconstruction is inaccurate during the low modulation index operation, due to the short time interval between two pulses’ edges. To improve this drawback, a method is proposed that generates an arbitrary voltage vector using Basic Voltage Vectors (BVVs) and collinear vectors, which guarantee the minimum time window for current measurement.

This paper deals with the development of the SVM method to reconstruct the three-phase load current by using a single-shunt measurement when low modulation index values are applied. Within the article, the new SVM method for low modulation index operation is presented. The SVM method is based on the collinear vector injection approach in order to solve the problem with the minimum time window needed for precise current measurement. The proposed method uses the single SVM pattern, which offers a simple implementation. Since the proposed SVM pattern introduces asymmetry, the exact sampling positions when average phase currents occur are derived. Section 2 describes the SVM principle applied for the low modulation index values, where a single SVM pattern is used. The method is based on the collinear vector injection approach, which ensures enough time for precise current measurement. Due to the introduced asymmetry because of the additional collinear vector injection, the current sampling position is not between the two SVM edges and changes based on the reference vector position. Section 3 derives exact sampling positions when the average phase current is measured in the sampling interval. Section 4 deals with the experimental verification of the proposed SVM method.

## 2. Current Reconstruction Method under the Low Modulation Index Inverter Operation

As mentioned above, the proposed method is applied for low modulation index values, where SVM and SVM-shift methods [25,26] cannot be applied because the times between the two SVM signal edges are too short. Figure 2 shows a vector diagram for a three-level inverter with a marked area, which is covered with the proposed method. The switching states and vector notation are indicated in Table 1, where x=1,2,3 and y=A,B,C. It can be seen that, instead of a hexagon, the shape that is formed from the BVV ends is a rectangle. The rectangle can be divided into four regions, as shown in Figure 2. Using the proposed method, an arbitrary voltage vector can be generated anywhere inside the rectangle shown in Figure 2. The maximum arbitrary voltage vector that can be generated using the proposed SVM method is shown in Figure 2 as ${\vec{V}}_{ref,max}$. This can be achieved when the theoretically maximum modulation index for the proposed SVM method is applied, which can be calculated as follows:(1)$$\begin{array}{c} m_{i,\;max}=\sqrt{3}\frac{U_{ref,max}}{U_{DC}}=0.289. \end{array}$$
To generate an arbitrary voltage vector ${\vec{V}}_{ref}$ as shown in Figure 2, the pattern based on SVM can be used (Figure 3a). It can be used to generate an arbitrary voltage vector, but when times $t_{1}/2$ or $t_{2}/2$ are shorter than $T_{min}$, then the precise current measurement cannot be achieved without additional modification due to the insufficient time widow needed for measurement. The time duration between the two edges is long enough if it is longer than $T_{min}$, which depends on the hardware design, dead time, and settling time [26], and can be evaluated as follows:(2)$$\begin{array}{c} T_{min}=T_{DT}+T_{PD}+T_{r}+T_{settling}+T_{S\&H}, \end{array}$$
where $T_{DT}$ is the dead time between the two triggering pulses of the two complementary MOSFETs in order to avoid a short-circuit, $T_{PD}$ is the gate driver propagation delay, $T_{r}$ is the rise time of the amplifier including the power switches’ (MOSFETs) turn on time, $T_{settling}$ is the settling time of amplifier when the measured signal stabilizes, and $T_{S\&H}$ is the sample and hold time of the A/D converter. This drawback occurs when a low modulation index is applied. One of the solutions can be introducing asymmetry to the SVM pattern by moving vectors ${\vec{V}}_{1}$ and ${\vec{V}}_{2}$ from the falling SVM signal edge to the rising SVM signal edge, as is shown in Figure 3b. Such an approach enables operation with a smaller modulation index compared to the SVM pattern shown in Figure 3a, but the method does not guarantee precise measurement when the interval $t_{1}$ or $t_{2}$ is still shorter than $T_{min}$.

To achieve inverter operation even with a smaller modulation index, the collinear vector injection approach is proposed to reach very low modulation index values ($m_{i}<0.1$), as is shown in Figure 4a–c. The SVM pattern consists of a zero voltage vector and four BVVs. BVVs ${\vec{V}}_{1}$ and ${\vec{V}}_{2}$ are considered as Regular Basic Voltage Vectors (RBVV), and BVV vectors ${\vec{V}}_{5}$ and ${\vec{V}}_{6}$ are considered as Injected Collinear Basic Voltage Vectors (ICBVVs).

Therefore, the time duration of the RBVVs ${\vec{V}}_{1}$ and ${\vec{V}}_{2}$ is prolonged for $T_{min}$ ($d_{min}=T_{min}/T_{s}$), ($t_{1}\to t_{1}+T_{min}$, and $t_{2}\to t_{2}+T_{min}$). Due to these inserts, the “wrong” reference voltage ${\vec{V}}_{ref1}$ is generated instead of the desired ${\vec{V}}_{ref}$, as shown in Figure 4b. To obtain the desired vector again, it is suggested that after the generation of ${\vec{V}}_{0}$ (Figure 4a), the collinear vectors ${\vec{V}}_{5}$ and ${\vec{V}}_{6}$ can be injected into the switching sequences, as shown in Figure 4c. The length of these vectors is adjusted by $d_{min}$. Furthermore, time $t_{0}$ needs to be reduced by $4T_{min}$ in order to keep the switching period unchanged. Therefore, all time intervals indicated in Figure 3 and Figure 4 are calculated as follows:(3)$$\begin{array}{c} t_{1}=d_{2}T_{s} \end{array}$$
(4)$$\begin{array}{c} t_{2}=d_{3}T_{s} \end{array}$$
(5)$$\begin{array}{c} t_{0}=T_{s}-t_{1}-t_{2}-4T_{min}. \end{array}$$
Using the length and position of the reference voltage vector ${\vec{V}}_{ref}$, represented in the α-β coordinate system, the length of vectors $d_{2}{\vec{V}}_{2}$ and $d_{3}{\vec{V}}_{3}$ (in Region 2, Figure 5) can be evaluated by calculating the duty cycle functions $d_{2}$ and $d_{3}$ as follows in the further analyses:(6)$$\begin{array}{c} {\vec{V}}_{ref}=\left[\begin{array}{c} |{\vec{V}}_{ref\alpha }|\\ |{\vec{V}}_{ref\beta }| \end{array}\right]=\left[\begin{array}{c} \cos\left(\omega t\right)\left|{\vec{V}}_{ref}\right|\\ \sin\left(\omega t\right)\left|{\vec{V}}_{ref}\right| \end{array}\right], \end{array}$$
where the directions α and β were chosen as the directions of vectors ${\vec{V}}_{1}({\vec{V}}_{13})$ and ${\vec{V}}_{8}$, as shown in Figure 2 and Figure 5, and $\omega t∊(\frac{\pi }{3},\frac{2\pi }{3})$. Vectors ${\vec{V}}_{2}$ and ${\vec{V}}_{3}$ are notated by coordinates as follows
(7)V→2=|V→2α||V→2β|=cos(ππ3)3)|V→2|sin(ππ3)3)|V→2|
(8)V→3=|V→3α||V→3β|=cos(ππ3)3)|V→2|−sin(ππ3)3)|V→2|therefore, an arbitrary voltage vector ${\vec{V}}_{ref}$ positioned inside Region 2 can be expressed as a linear combination of two RBVVs ${\vec{V}}_{2}$ and ${\vec{V}}_{3}$ and zero vector ${\vec{V}}_{0}$ (with a length of zero) as follows:
(9)$$\begin{array}{c} {\vec{V}}_{ref}=d_{2}{\vec{V}}_{2}+d_{3}{\vec{V}}_{3}+d_{0}{\vec{V}}_{0} \end{array}$$and further, combining (7) to (9), it can be obtained that:(10)$$\begin{array}{c} \left[\begin{array}{c} \cos\left(\omega t\right)\left|{\vec{V}}_{ref}\right|\\ \sin\left(\omega t\right)\left|{\vec{V}}_{ref}\right| \end{array}\right]=d_{2}\left[\begin{array}{c} \frac{1}{2}\left|{\vec{V}}_{2}\right|\\ \frac{\sqrt{3}}{2}\left|{\vec{V}}_{2}\right| \end{array}\right]+d_{3}\left[\begin{array}{c} \frac{1}{2}\left|{\vec{V}}_{3}\right|\\ -\frac{\sqrt{3}}{2}\left|{\vec{V}}_{3}\right| \end{array}\right]. \end{array}$$Considering that $\left|{\vec{V}}_{2}\right|=\left|{\vec{V}}_{3}\right|=\frac{1}{3}U_{DC}$ and manipulating with (2), it is obtained that:(11)$$\begin{array}{ccc} & & d_{2}=\frac{3|{\vec{V}}_{ref}|}{U_{DC}}\left(\cos\left(\omega t\right)+\frac{1}{\sqrt{3}}\sin\left(\omega t\right)\right)\\ & & d_{3}=\frac{3|{\vec{V}}_{ref}|}{U_{DC}}\left(\cos\left(\omega t\right)-\frac{1}{\sqrt{3}}\sin\left(\omega t\right)\right). \end{array}$$As can be seen from (11), duty cycle functions *d*_2_ and *d*_3_ can be positive or negative in different regions. Therefore, depending on the reference voltage vector position, this fact could be applied for the introduction of the collinear vectors into the modulation procedure. If duty cycle *d*_2_ is positive, then the RBVV is used as BVV ${\vec{V}}_{2}$ and the ICBVV is used as BVV ${\vec{V}}_{5}$. If duty cycle *d*_2_ is negative, then the RBVV is used as BVV ${\vec{V}}_{5}$ and the CIBVV is used as BVV ${\vec{V}}_{2}$. In an analogous way, based on duty cycle *d*_3_, it is determined whether BVVs ${\vec{V}}_{3}$ and ${\vec{V}}_{6}$ are used as regular or injected BVVs. The described algorithm is presented with the flowchart diagram shown in Figure 6, where the duty cycle decision for SVM for low modulation index values can be seen. If both duty cycles are positive, the reference voltage vector is positioned inside Region 2. In such a case, calculated duty cycles *d*_2_ and *d*_3_ are increased for duty cycle *d*_*min*_ and are used as the duty cycles for the RBVV (vectors ${\vec{V}}_{2}$ and ${\vec{V}}_{3}$). The duty cycles for the CIBVV were set to *d*_*min*_, as described previously. If, for example, duty cycle *d*_2_ is negative and *d*_3_ is positive, the reference voltage vector is positioned inside Region 3. In such a case, RBVVs are used as vectors ${\vec{V}}_{5}$ and ${\vec{V}}_{3}$. Duty cycle *d*_2_ is increased for duty cycle *d*_*min*_ and is used as the duty cycle for RBVV ${\vec{V}}_{5}$. In a similar way, duty cycle *d*_3_ is used for generating RBVV ${\vec{V}}_{3}$.

It can be concluded that regions differ based on the regular and injected basic voltage vectors. For Region 1, vectors ${\vec{V}}_{2}$ and ${\vec{V}}_{6}$ are used as RBVVs and vectors ${\vec{V}}_{3}$ and ${\vec{V}}_{5}$ are used as CIBVVs. In a similar way, vectors ${\vec{V}}_{2}$ and ${\vec{V}}_{3}$ are used as RBVVs inside Region 2, vectors ${\vec{V}}_{3}$ and ${\vec{V}}_{5}$ are used inside Region 3, and vectors ${\vec{V}}_{5}$ and ${\vec{V}}_{6}$ are used inside Region 4. As CIBVVs, vectors ${\vec{V}}_{5}$ and ${\vec{V}}_{6}$ are used inside Region 2, vectors ${\vec{V}}_{2}$ and ${\vec{V}}_{6}$ inside Region 3, and vectors ${\vec{V}}_{2}$ and ${\vec{V}}_{3}$ inside Region 4.

## 3. Current Sampling Positions

The single SVM pattern is used for all regions, as shown in Figure 7. With the proposed single SVM pattern, low modulation index values can be achieved and enough time for current measurement is ensured in the whole presented rectangle area. Exact sampling positions need to be derived for precise current measurement. With the proposed SVM pattern shown in Figure 7, during the time interval when the vectors ${\vec{V}}_{2}$ and ${\vec{V}}_{5}$ are applied, the current that flows through the third phase $i_{w}\left(t\right)$ can be measured. During the time interval when the vectors ${\vec{V}}_{3}$ and ${\vec{V}}_{6}$ are applied, the current that flows through the second phase $i_{v}\left(t\right)$ can be measured.

Current waveforms for $i_{v}\left(t\right)$ and $i_{w}\left(t\right)$ shall be determined in order to calculate the exact sampling position. As an example, while the reference voltage vector is positioned inside the first region, the characteristic voltages and phase currents are shown in Figure 8. According to the second Kirchhoff law, the phase voltage can be determined as follows:(12)$$\begin{array}{c} u_{xn}\left(t\right)=u_{Lx}\left(t\right)+u_{Rx}\left(t\right), \end{array}$$
where x=u,v,w, $u_{Lx}$, and $u_{Rx}$ are load inductor and resistor voltages, respectively. The average value of the phase voltage is equal to the average value of the voltage across the resistor:(13)$$\begin{array}{c} U_{xn,avg}=U_{Rx,avg}. \end{array}$$
The average value of the phase voltage can be calculated as:(14)$$\begin{array}{c} U_{xn,avg}=\frac{1}{T_{s}}\int_{0}^{T_{s}}u_{xn}\left(t\right)\;dt \end{array}$$
which gives the average phase voltage, while an arbitrary voltage vector is positioned inside Region 1, for the second and third phases:(15)$$\begin{array}{c} U_{vn,avg}=\frac{1}{6}\frac{1}{T_{s}}U_{DC}\left(t_{1}-2t_{2}\right)=U_{Rv,avg} \end{array}$$
(16)$$\begin{array}{c} U_{wn,avg}=\frac{1}{6}\frac{1}{T_{s}}U_{DC}\left(t_{2}-2t_{1}\right)U_{Rw,avg}. \end{array}$$
From (12) and (13), the inductor voltage can be determined as follows:(17)$$\begin{array}{c} u_{Lx}\left(t\right)=u_{xn}\left(t\right)-U_{Rx,avg}. \end{array}$$
The waveform for the inductor voltages for the second $u_{Lv}\left(t\right)$ and third phases $u_{Lw}\left(t\right)$, while an arbitrary voltage vector is positioned inside Region 1, are shown in Figure 8. Knowing the inductor voltages $u_{Lv}\left(t\right)$ and $u_{Lw}\left(t\right)$, the current ripple for the phase currents for each time interval, shown in Figure 8, can be determined as follows:(18)$$\begin{array}{c} \Delta i_{xy}=\frac{u_{Lx}\cdot \Delta t}{L_{x}}, \end{array}$$
where *x* can be *u*, *v*, or *w* and *y* represents the time interval and can be any number from 1–7. Therefore, the phase current average value can be obtained:(19)$$\begin{array}{c} I_{x,avg}=\frac{1}{T_{s}}\int_{t_{0x}}^{t_{7x}}i_{x}\left(t\right)dt, \end{array}$$
where $T_{s}=t_{7x}-t_{0x}$, as indicated in Figure 8. According to the waveform for the second phase current $i_{v}\left(t\right)$, the current ripple can be determined as follows:(20)$$\begin{array}{c} \Delta i_{v}=\Delta i_{v2}+\Delta i_{v3} \end{array}$$
or:(21)$$\begin{array}{c} \Delta i_{v}=\Delta i_{v1}+\Delta i_{v4}+\Delta i_{v5}+\Delta i_{v6}+\Delta i_{v7}. \end{array}$$
Precise current measurement can be performed at the exact time instant when the average value occurs. The average value ($I_{v,avg}$) of phase current $i_{v}\left(t\right)$ can be measured during Intervals 2 and 5, while basic voltage vectors ${\vec{V}}_{3}$ and ${\vec{V}}_{6}$ are active. The first possible sampling position is while BVV ${\vec{V}}_{3}$ is active. Based on (18) and using (20) the current ripple, this can be evaluated as follows:(22)$$\begin{array}{ccc} \Delta i_{v} & = & \frac{1}{L_{v}}\left[\left(\frac{2}{3}U_{DC}-U_{Rv,avg}\right)T_{min}\right.\\ & + & \left.\left(\frac{1}{3}U_{DC}-U_{Rv,avg}\right)\left(T_{min}+t_{1}\right)\right]. \end{array}$$
Based on calculated time $t_{1}$ using (3) and (11) and from (22), it can be concluded that the current that flows through the second phase $i_{v}\left(t\right)$ never reaches the average value $I_{v,avg}$ while basic voltage vector ${\vec{V}}_{3}$ is active. This will be true if the following condition is fulfilled:(23)$$\begin{array}{c} \Delta i_{v2}<\Delta i_{v3}. \end{array}$$
Therefore, according to the second current sampling position, which is suggested and evaluated in a similar way as for the first sampling position, it can be determined whether the average phase current occurs or not while basic voltage vector ${\vec{V}}_{6}$ is active. Since:(24)$$\begin{array}{c} \Delta i_{v1}=\Delta i_{v7}=\frac{\Delta i_{v4}}{2} \end{array}$$
and due to the symmetry, where $\Delta i_{v4}$ occurs above average current $I_{v,avg}$ and ($\Delta i_{v1}$ + $\Delta i_{v7}$) occurs bellow average current $I_{v,avg}$, to determine whether the average phase current measurement is possible while basic voltage vector ${\vec{V}}_{6}$ is active, according to (24), it is sufficient to observe only the ripples $\Delta i_{v5}$ and $\Delta i_{v6}$ where average current $I_{v,avg}$ will occur, which gives:(25)$$\begin{array}{c} \Delta i_{v56}=\Delta i_{v5}+\Delta i_{v6}. \end{array}$$
The average current will occur when current ripple (25) is equal to half of the value ($\Delta i_{v,avg}=\Delta i_{v56}/2$) as follows:(26)$$\begin{array}{c} \Delta i_{v,avg}=\frac{1}{2}\left(\Delta i_{v5}+\Delta i_{v6}\right). \end{array}$$
Combining (18) and (26), it is obtained that:(27)$$\begin{array}{ccc} \Delta i_{v,avg} & = & \frac{1}{2}\frac{1}{L_{v}}\left[\left(\frac{2}{3}U_{DC}-U_{Rv,avg}\right)\left(T_{min}+t_{2}\right)\right.\\ & + & \left.\left(\frac{1}{3}U_{DC}-U_{Rv,avg}\right)T_{min}\right]. \end{array}$$
Since the condition:(28)$$\begin{array}{c} \left.\left(\frac{2}{3}U_{DC}-U_{Rv,avg}\right)\left(T_{min}+t_{2}\right)\right.\\ >\left.\left(\frac{1}{3}U_{DC}-U_{Rv,avg}\right)T_{min}\right., \end{array}$$
is always true, it can be concluded that it is possible to perform the second current measurement when basic voltage vector ${\vec{V}}_{6}$ is active, all the time, while the reference voltage vector is positioned inside Region 1. The average current value $I_{v,avg}$ in the fifth time interval (Figure 8) occurs when current ripple $\Delta i_{v5}$ is equal to (26), as follows:(29)$$\begin{array}{c} \Delta i_{v,avg}=\frac{1}{L_{v}}\left(\frac{2}{3}U_{DC}-U_{Rv,avg}\right)t_{m3}. \end{array}$$

Combining (27) and (29), it yields:(30)$$\begin{array}{c} \left.\frac{1}{L_{v}}\left(\frac{2}{3}U_{DC}-U_{Rv,avg}\right)t_{m3}=\right.\\ \left.\frac{1}{2}\frac{1}{L_{v}}[\left(\frac{2}{3}U_{DC}-U_{Rv,avg}\right)\left(T_{min}+t_{2}\right)\right.\\ +\left.\left(\frac{1}{3}U_{DC}-U_{Rv,avg}\right)T_{min}\right.]. \end{array}$$
Since (15) can be included in (30), the exact time $t_{m3}$ when average current $I_{v,avg}$ occurs can be calculated and is indicated in Table 2 for Region 1. With the same approach, exact sampling positions for phase current $i_{w}\left(t\right)$ can be determined, as also indicated in Table 2. The average current for the third phase can always be measured while BVV ${\vec{V}}_{2}$ is active and while the reference voltage vector is positioned inside Region 1. In the case when BVV ${\vec{V}}_{5}$ is active, it could happen that the average current $I_{w,avg}$ does not occur in the sixth time interval. Because of this, the third phase current is measured while BVV ${\vec{V}}_{2}$ is active. With the same approach presented for the first region, for exact sampling positions, when the average current can be measured is also derived for the other regions and indicated in Table 2. Figure 7 shows the exact sampling positions extracted from Table 2. In these time instants, the current samples are taken and the average phase current has occurred.

## 4. Experimental Results

The designed three-level inverter, with three-phase R-L load with 1 Ω and 560 μH, was tested under laboratory conditions. The test-bench system is shown in Figure 9. The DC-link voltage $U_{DC}$, the switching frequency $f_{s}$, and the minimum time window $T_{min}$ were considered as 24 V, 16 kHz, and 4.5 μs, respectively. The single-shunt 5 mΩ was placed in the DC-link, as shown in Figure 1, and the current measurement circuit was designed for a 16 A maximum current measurement range. A Texas Instruments control board based on TMS320F28335 was used to implement the proposed method. The proposed method was designed using the model-based design approach using MATLAB/Simulink with a measured CPU load of 25 μs (40%) including the additional debugging functionalities. The designed algorithm can be further optimized, and the CPU load can be reduced. Floating-point or fix-point DSP, with enough PWM outputs and fast ADC with the flexible triggering and synchronization options, can be used for the algorithm implementation. Through serial communication, reconstructed currents were obtained using an X2C Scope within the runtime. To verify the single-shunt current reconstruction measurement results, these were compared with the reference currents measured with the oscilloscope.

Figure 10 and Figure 11 present the reconstruction results for the low modulation index operation $m_{i}=0.05$ and 0.075, respectively. As a reference measured signal, the current signals obtained by the three Hall-sensors and the oscilloscope were used. The measurements were performed at four different load frequency requirements, at 25 Hz, 50 Hz, 75 Hz, and 100 Hz, respectively.

Due to the introduced asymmetry, because of the additionally injected collinear vectors, an increased current ripple can be observed for the two phase currents $i_{v}\left(t\right)$ and $i_{w}\left(t\right)$. Even with asymmetric SVM signals and increased current ripple, due to the derived expressions for the exact sampling position when the average phase current occurs, the phase currents were reconstructed within a 5 % measurement error, as shown in Table 3. The measurement error is calculated as follows:(31)$$\begin{array}{c} \epsilon =|\frac{I_{f-peak}-I_{f-peak,rec}}{I_{f-peak}}|\cdot 100 \end{array}$$
where $I_{f-peak}$ represents the measured average peak phase current value (by the Hall sensor and scope) and $I_{f-peak,rec}$ represents the reconstructed peak phase current value. With a higher modulation index, the measurement error was reduced, due to the current measurement range, which was designed for a maximum 16 A. Furthermore, it can be observed that the occurrence of current spikes was removed from the reconstructed signals. Figure 12 verifies the proposed algorithm behavior under the transient condition state. Figure 12a shows the case when modulation index changes leap from $m_{i}=0.075$ to $m_{i}=0.15$, while the load frequency and DC-link voltage were 100 Hz and 24 V, respectively. Figure 12b shows the case when DC-link voltage changes leap from $U_{DC}=24\;V$ to $U_{DC}=48\;V$, while the load frequency and modulation index were 10 Hz and 0.075, respectively. The experiment (Figure 12) confirmed the proposed SVM algorithm’s operation under the transient conditions.

## 5. Conclusions

The goal of the presented reconstruction method was to develop the SVM method for a low modulation index converter operation when the single-shunt current measurement signal was applied to a three-level inverter. The SVM method using the collinear vector injection approach and based on a single SVM pattern was developed, in order to cover the low modulation index operation. Due to the introduced asymmetry, the exact sampling position when average phase current occurs was obtained analytically. The proposed SVM method was verified through an experimental test-bench. The proposed SVM method offers an appropriate solution for low modulation index values where the SVM and SVM shift methods cannot be applied due to the insufficient time window needed for precise current measurement. The three-level inverter operation with a modulation index less than 0.2 is achievable, and single-shunt current reconstruction is possible to perform. The proposed method can be used as part of a hybrid solution, together with the SVM shift method. As a disadvantage, due to the asymmetric SVM pattern, the current ripple increased.

## Figures and Tables

**Figure 1 sensors-22-02249-f001:**
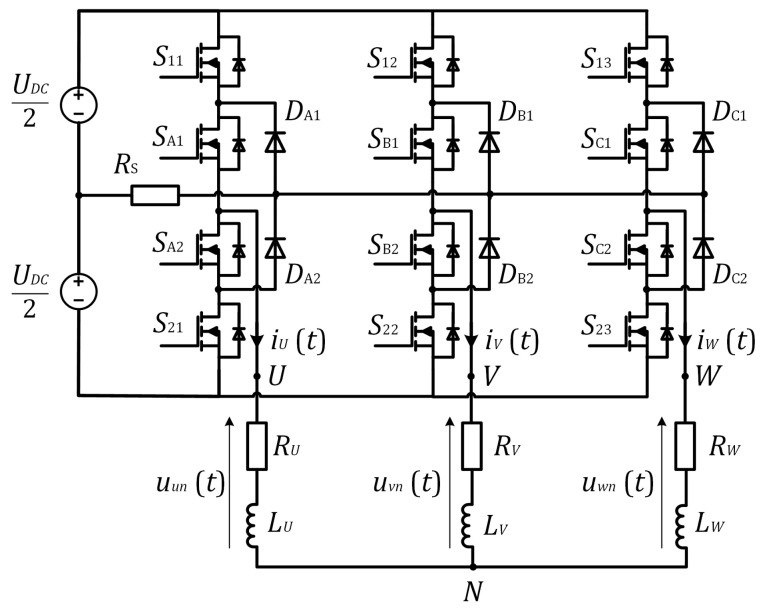
Three-level DC–AC converter.

**Figure 2 sensors-22-02249-f002:**
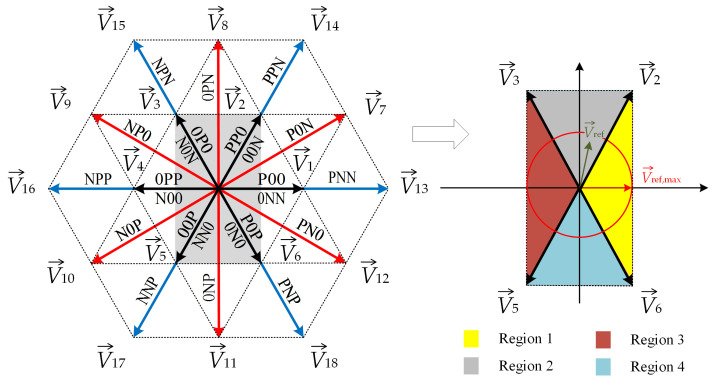
Vector diagram for the three-level inverter.

**Figure 3 sensors-22-02249-f003:**
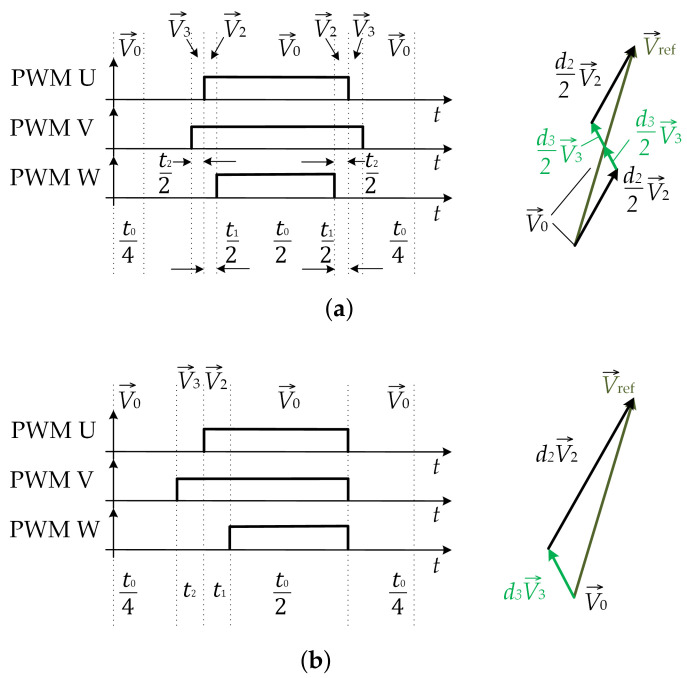
SVM pattern for Region 2: (**a**) using ordinary SVM; (**b**) using vector shifting to the rising edge.

**Figure 4 sensors-22-02249-f004:**
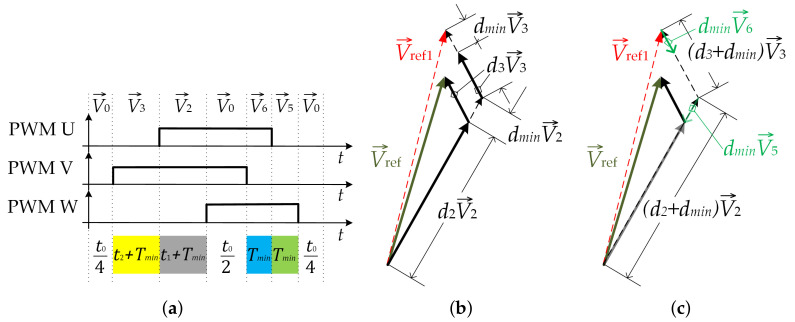
SVM pattern for Region 2: (**a**) vectors versus time intervals; (**b**) injection of vectors $d_{min}{\vec{V}}_{2}$ and $d_{min}{\vec{V}}_{3}$; (**c**) injection of collinear vectors $d_{min}{\vec{V}}_{5}$ and $d_{min}{\vec{V}}_{6}$.

**Figure 5 sensors-22-02249-f005:**
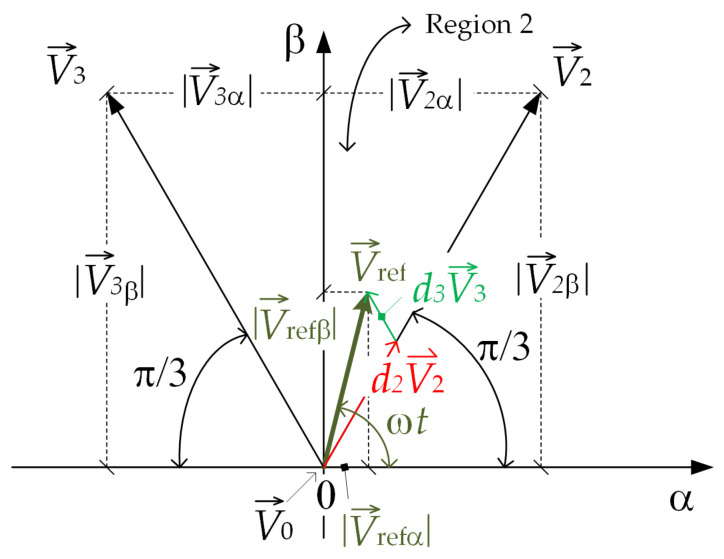
Reference and basic vectors’ components.

**Figure 6 sensors-22-02249-f006:**
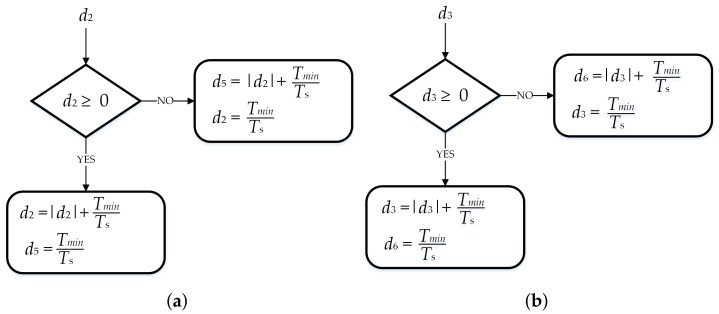
Flowchart diagram for the duty cycle decision when SVM is used for low modulation index operation: (**a**) for $d_{2}$; (**b**) for $d_{3}$.

**Figure 7 sensors-22-02249-f007:**
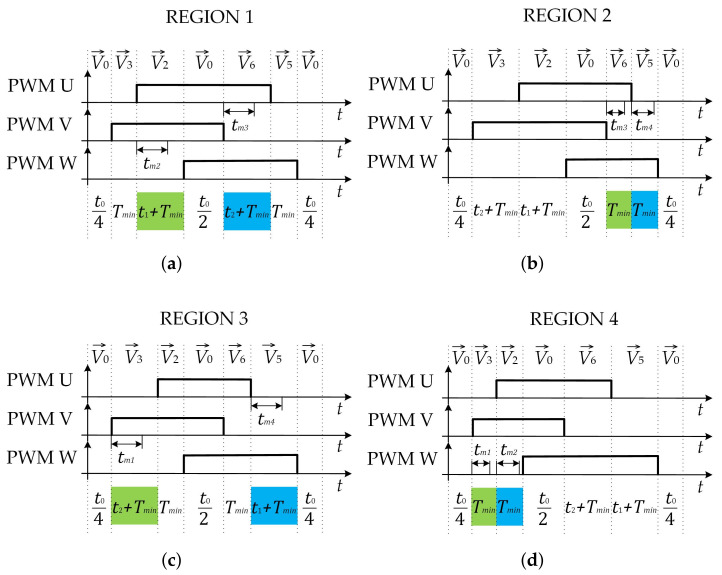
Proposed SVM pattern for low modulation index values: (**a**) Region 1; (**b**) Region 2; (**c**) Region 3; (**d**) Region 4.

**Figure 8 sensors-22-02249-f008:**
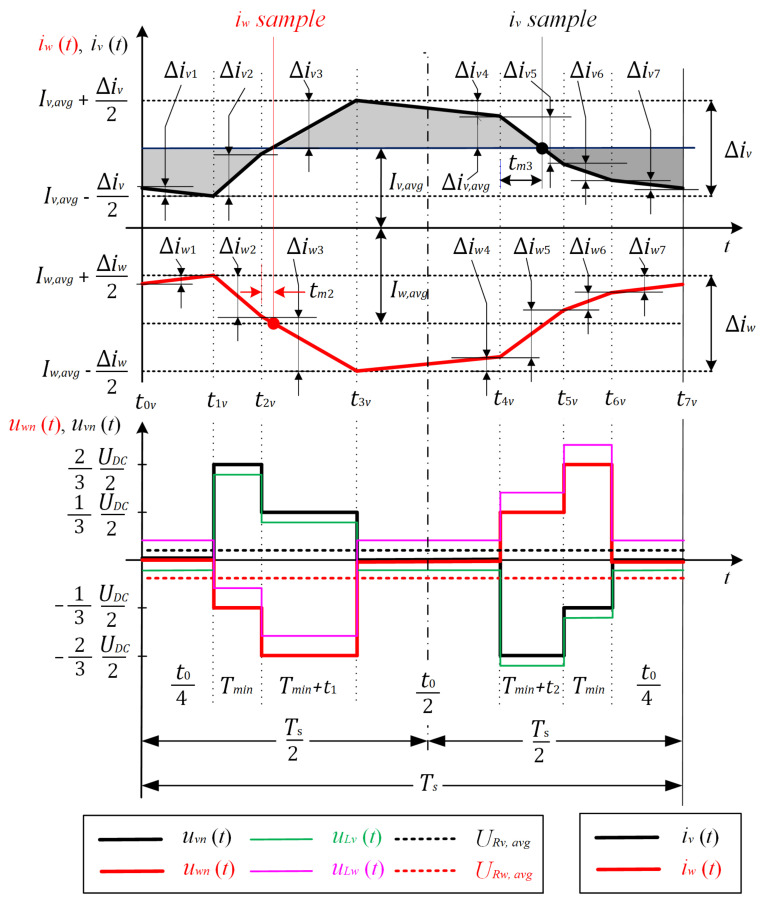
Characteristic voltages and phase currents for Region 1.

**Figure 9 sensors-22-02249-f009:**
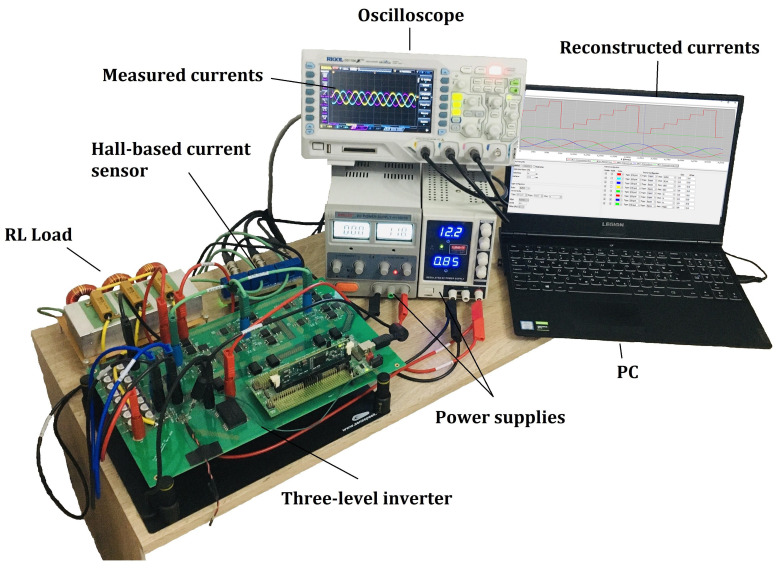
Measurement test-bench.

**Figure 10 sensors-22-02249-f010:**
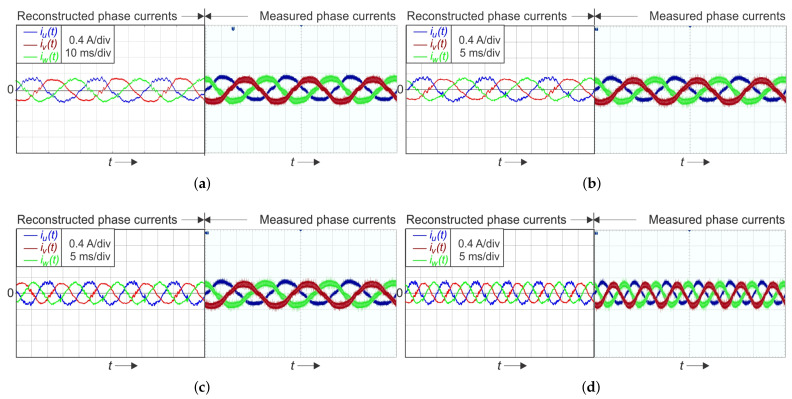
Single-shunt reconstruction of a three-phase current using SVM for low modulation index $m_{i}=0.050$ at a current magnitude of ${\hat{i}}_{u}={\hat{i}}_{v}={\hat{i}}_{w}=0.3\;A$: (**a**) reconstructed and oscilloscope-measured currents at f=25Hz; (**b**) at f=50Hz; (**c**) at f=75Hz; (**d**) at f=100Hz.

**Figure 11 sensors-22-02249-f011:**
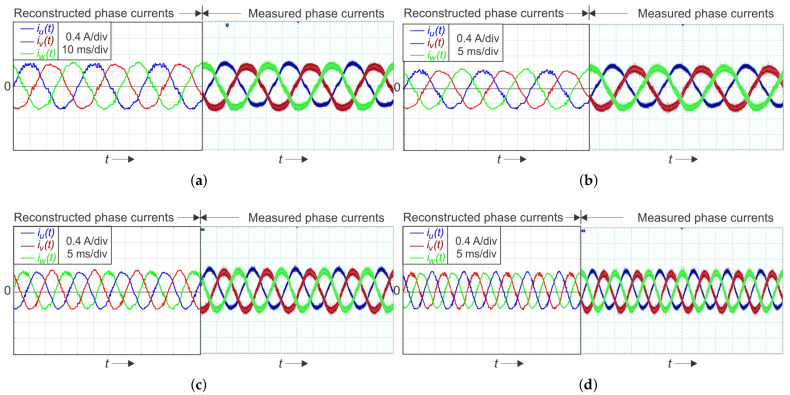
Single-shunt reconstruction of a three-phase current using SVM for low modulation index $m_{i}=0.075$ at a current magnitude of ${\hat{i}}_{u}={\hat{i}}_{v}={\hat{i}}_{w}=0.5\;A$: (**a**) reconstructed and oscilloscope-measured currents at f=25Hz; (**b**) at f=50Hz; (**c**) at f=75Hz; (**d**) at f=100Hz.

**Figure 12 sensors-22-02249-f012:**
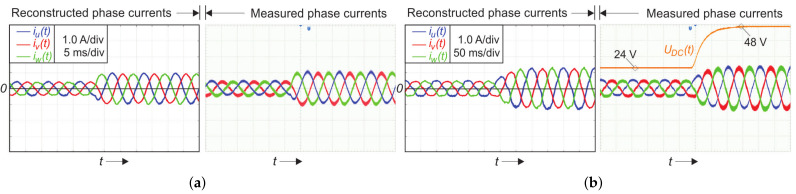
Single-shunt reconstruction of a three-phase current using SVM for a low modulation index under the transient condition: (**a**) modulation index step from $m_{i}=0.075$ to $m_{i}=0.15$ at f=100Hz; (**b**) DC-link voltage leap from $U_{DC}=24\;V$ to $U_{DC}=48\;V$ at f=10Hz and $m_{i}=0.075$.

**Table 1 sensors-22-02249-t001:** The switching state explanation.

$S_{1x}$	$S_{y1}$	$S_{y2}$	$S_{2x}$	$U_{x0}$	Notation
1	1	0	0	$U_{DC}/2$	P
0	1	1	0	0	0
0	1	1	1	$-U_{DC}/2$	N

**Table 2 sensors-22-02249-t002:** Expression for precise sampling position calculation.

Region	Expression
Region 1	$t_{m2}=\frac{1}{2}\left(t_{1}+T_{min}\right)-\frac{1}{2}T_{min}\left(\frac{T_{s}-2t_{1}+t_{2}}{2T_{s}-2t_{1}+t_{2}}\right)$
	$t_{m3}=\frac{1}{2}\left(t_{2}+T_{min}\right)+\frac{1}{2}T_{min}\left(\frac{T_{s}+t_{1}-2t_{2}}{2T_{s}+t_{1}-2t_{2}}\right)$
Region 2	$t_{m3}=\frac{1}{2}T_{min}+\frac{1}{2}T_{min}\left(\frac{T_{s}+t_{1}+2t_{2}}{2T_{s}+t_{1}+2t_{2}}\right)$
	$t_{m4}=\frac{1}{2}T_{min}-\frac{1}{2}T_{min}\left(\frac{T_{s}+2t_{1}+t_{2}}{2T_{s}+2t_{1}+t_{2}}\right)$
Region 3	$t_{m1}=\frac{1}{2}\left(t_{2}+T_{min}\right)+\frac{1}{2}T_{min}\left(\frac{T_{s}+t_{1}-2t_{2}}{2T_{s}+t_{1}-2t_{2}}\right)$
	$t_{m4}=\frac{1}{2}\left(t_{1}+T_{min}\right)-\frac{1}{2}T_{min}\left(\frac{T_{s}-2t_{1}+t_{2}}{2T_{s}-2t_{1}+t_{2}}\right)$
Region 4	$t_{m1}=\frac{1}{2}T_{min}+\frac{1}{2}T_{min}\left(\frac{T_{s}+t_{1}+2t_{2}}{2T_{s}+t_{1}+2t_{2}}\right)$
	$t_{m2}=\frac{1}{2}T_{min}-\frac{1}{2}T_{min}\left(\frac{T_{s}+2t_{1}+t_{2}}{2T_{s}+2t_{1}+t_{2}}\right)$

**Table 3 sensors-22-02249-t003:** Performance index for the measurement results.

Frequency (Hz)	Modulation Index $m_{i}$	Reconstructed Current $I_{f-\mathrm{peak},\mathrm{rec}}\;$(A)	Scope Measurement $I_{f-\mathrm{peak}}$ (A)	Relative Error (%)
25	0.05	0.302	0.287	5.23
	0.075	0.526	0.512	2.73
50	0.05	0.293	0.284	3.17
	0.075	0.517	0.504	2.58
75	0.05	0.285	0.272	4.78
	0.075	0.499	0.488	2.25
100	0.05	0.276	0.263	4.94
	0.075	0.490	0.480	2.08

## Data Availability

Data are contained within the article at hand.

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
