# Peer review of "Single-Shunt Measurement of Three-Phase Currents for a Three-Level Inverter under the Low Modulation Index Operation"

_sensors, 2022, doi:10.3390/s22062249_

Round 1
Reviewer 1 Report
This paper deals with three-phase current reconstruction method under the low modulation index operation of three-phase three-level PWM inverters by using the single-shunt current signal measurement. The shunt is placed in a dc-link. The proposed reconstruction method is based on the combination of collinear vector injection and shifting of Space Vector Modulation (SVM) signals. The method offers a good solution for the area where the small modulation index appears. In this case, the lack of measurement time exists due to the narrow triggering pulses. This approach was studied theoretically and verified by laboratory experiments.
- For the full text of 14 pages, 21 references are a bit small. Moreover, all the references are located in the introduction section, and there is not a single reference in the main part of the article, which is very unreasonable.
- In the introduction part, the article introduces the research status of the subject too much space, but basically lacks the research background, purpose and significance of the subject. The innovation points of the article are not clearly described, which may be difficult for readers to understand.
- References, citation images, and some formula labels use blue, which is counterintuitive. Since the image of the article contains a lot of blue marks, the blue marks appearing in the article can easily mislead readers and affect the beauty of the entire article.
- There are still some problems with the details of the article, such as the format of Forms 1,2 and 3 is different from the usual, but in a strange way. Also, images 3 and 7 are not centered, and images 10 and 11 are too far to the right. All of the above will affect the reader's reading experience, and by modifying these details, the article can be more beautiful.
- The entire article, including the references, is marked with the number of lines on the far left, which is not conducive to the beauty of the overall structure of the article and may even cause misunderstandings to readers. It is recommended to remove these line number annotations.
- Picture problems in the article. Excessive colors are included in b, c in Figure 5 and in Figure 8, which may affect the overall aesthetics of the image and the reader's understanding due to the addition of too many tags to these images. Also, the clarity of the photo in Figure 9 is a bit low, and these issues need to be paid attention to later.
- There are also some irregularities in the format in the article. The initial word "Where" for the interpretation of variables in the article should be top case and lowercase; the interval between lines of some chapters is not uniform. By modifying these details, the article can be more beautiful.
- The conclusion part of the article is a bit simple, there is no summary and review of the full-text work, and there is also a lack of some summary and outlook for the research topic.
Author Response
We have followed all of your suggestions.

Reviewer 2 Report
- The introduction section must be improved. More existing approaches should be explained in intorduction.
- Compare your approach to existing ones.
Author Response

(The authors gave the same response as above.)

Reviewer 3 Report
The authors presented the results of scientific investigation on a specific method of control, implemented in the DC/AC inverters. The work is devoted to the presentation and validation of the proposed algorithm, taking into account low modulation index of the signals.
Some different states of the method are discussed. The presented method was derived using general rules, and typical vector representation of the signals. The validation of the proposed scheme was carried out on the basis of the selected system, however, the presented results and the text of the paper contain some elements of generalization. In this way, the presented results can be useful for other researchers and can be implemented to other systems.
I assess correctly the methodology of the research work. The authors presented the analytical model of the proposed method, studied the different states. The results of numerical simulations were verified using results of measurements on a laboratory stand.
The general scheme of the paper is correct. The discussion of the other works is rather short. It can be improved, and extended (particularly to explain pros of some other algorithms. The implemented notation is correct and consistent. The language of the paper is correct. The article requires some small corrections. Detailed remarks regarding errors in notation, form of text, and wording are marked in the attached file.
Below I present some critical comments on the results of the work.
- What is the dynamics of the algorithm when some transient states can be occurred.
- The described algorithm can be implemented using an embedded, dedicated computer system. What is the technical requirements for the computer (computational performance, latency, etc.)?
My score of implications for research and practice goes to the average, but acceptable rank. In my opinion the conclusions are too general (no quantitative analysis).

Author Response

(The authors gave the same response as above.)

Round 2
Reviewer 1 Report
After revision, the manuscript could be accepted.
Reviewer 2 Report
No additional comments